# Deep Reasoning Networks for Unsupervised Pattern De-mixing

## Abstract

We introduce Deep Reasoning Networks (DRNets), an end-to-end framework that combines deep learning with reasoning for solving pattern de-mixing problems, typically in an unsupervised or weakly-supervised setting. DRNets exploit problem structure and prior knowledge by tightly combining logic and constraint reasoning with stochastic-gradient-based neural network optimization. We illustrate the power of DRNets on de-mixing overlapping hand-written Sudokus (Multi-MNIST-Sudoku) and on a substantially more complex task in scientific discovery that concerns inferring crystal structures of materials from X-ray diffraction data (Crystal-Structure-Phase-Mapping). DRNets significantly outperform the state of the art and experts' capabilities on Crystal-Structure-Phase-Mapping, recovering more precise and physically meaningful crystal structures. On Multi-MNIST-Sudoku, DRNets perfectly recovered the mixed Sudokus' digits, with 100% digit accuracy, outperforming the supervised state-of-the-art MNIST de-mixing models.

## 1 Introduction

Deep learning has achieved tremendous success in areas such as vision, speech recognition, language translation, and autonomous driving. Nevertheless, certain limitations of deep learning are generally recognized, in particular, limitations due to the fact that deep learning approaches heavily depend on the availability of large amounts of labeled data. In certain domains, such as scientific discovery, it is often the case that scientists don't have large amounts of labeled data and instead have to rely on prior knowledge to make sense of the data. One grand challenge in scientific discovery is to perform high-throughput unsupervised interpretation of scientific data, given its exponential growth in generation rates, dramatically outpacing humans' ability to analyze them. Herein we consider pattern de-mixing problems, which involve decomposing a mixed signal into the collection of source patterns, such as separating mixtures of X-ray diffraction (XRD) signals into the source XRD signals of the corresponding crystal structures, a key challenge in materials discovery. More generally, pattern de-mixing problems are pervasive in scientific areas as diverse as biology, astronomy, and materials science, as well as in commercial applications for e.g., healthcare and music.

We propose **Deep Reasoning Networks (DRNets)**, an end-to-end framework that combines deep learning with logical and constraint reasoning for solving unsupervised or very-weakly-supervised pattern de-mixing tasks. We illustrate the power of DRNets for disentangling two overlapping hand-written Sudokus (**Multi-MNIST-Sudoku**) (see Fig.1) and for solving a substantially more complex de-mixing task in scientific discovery that concerns inferring crystal structures of materials from X-ray diffraction data, which we refer to as **Crystal-Structure-Phase-Mapping**. Both de-mixing tasks require probabilistic reasoning to interpret noisy and uncertain data, while satisfying a set of rules: Sudoku rules and thermodynamic rules, respectively. For example, de-mixing hand written digits is challenging, but it becomes more feasible when we reason about the prior knowledge concerning the two overlapping Sudokus. Crystal structure phase mapping is yet substantially more complex. In fact, crystal structure phase mapping easily becomes too complex for experts to solve and is a major bottleneck in high-throughput materials discovery. DRNets are inspired and motivated by problems from scientific discovery, such as crystal structure phase mapping.

**Our contributions: (1)** We introduce **Deep Reasoning Networks (DRNets)**, an end-to-end framework that combines deep learning with logical and constraint reasoning for unsupervised or very-weakly-supervised de-mixing tasks. Specifically, DRNets perform end-to-end deep reasoning by

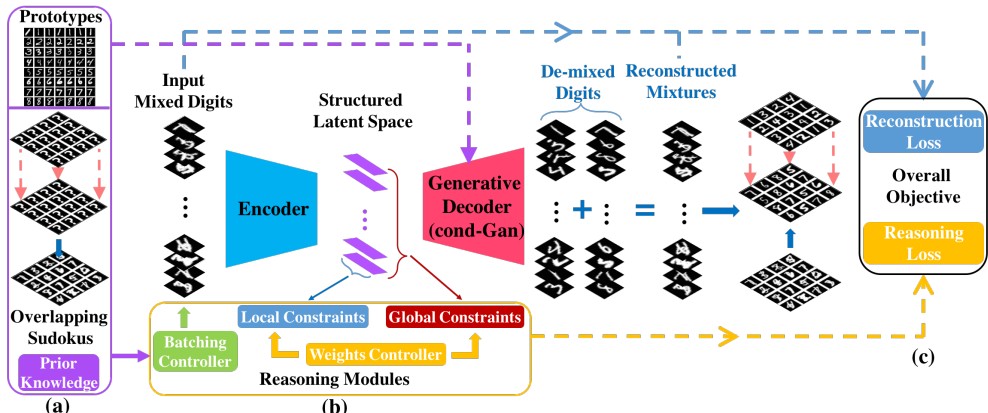

**(a) The two ground-truth Sudokus**  **(b) Input mixture**  **(c) DRNet de-mixed Sudokus**  **(d) Reconstructed mixture**

Figure 1: **(a)** Two 4x4 Sudokus: The cells in each row, column, and any of the four 2x2 boxes involving the corner cells have non-repeating digits. **(b)** Two overlapping Sudokus, with a mixture of two digits in each cell: one from 1 to 4 and the other from 5 to 8. In **Multi-MNIST-Sudoku**, the digits of two overlapping hand written Sudokus (b) have to be de-mixed (as done by DRNets in **(c)**). **(d)** The reconstructed overlapping hand written Sudokus from DRNets.

Figure 2: Deep Reasoning Networks (DRNets) perform end-to-end deep reasoning by encoding a latent space of the input data that captures prior knowledge constraints and is used by a generative decoder to generate the targeted output. **(a)** Prior knowledge includes prototypes of digits, which are used to pre-train and build the decoder's generative module, and Sudoku's rules, which help DRNet reason about the overlapping digits. **(b)** Reasoning modules batch data points involved in the same constraints (cells in rows, columns, blocks of a Sudoku) together, enforce that the structure of the latent space satisfies prior knowledge, and dynamically adjust the weights of constraints based on their satisfiability. **(c)** The overall objective combines responses from the generative decoder (thinking fast) and the reasoning modules (thinking slow).

encoding a latent space of the input data that captures the structure and prior knowledge constraints within and among data points (Fig.2). The latent space is used by a generative decoder to generate the targeted output, which should be consistent with the input data and prior knowledge. Subsequently, DRNets optimize an objective function capturing the overall problem objective as well as prior knowledge in the form of weighted constraints. **(2)** To instantiate the logical constraints in DRNets, we introduce a group of **entropy-based continuous relaxations** that use probabilistic modeling to encode general discrete constraints including sparsity, cardinality and so-called All-Different constraints. To optimize those constraints, we introduce a variant of standard SGD method (Robbins & Monro, 1985) called constraint-aware stochastic gradient descent, which batches data points involved in the same constraint component together and dynamically adjust the constraints' weights as a function of their satisfiability. In the following sections, we show how to encode Multi-MNIST-Sudoku and Crystal-Structure-Phase-Mapping as DRNets, by properly defining the structure of the latent space, additional reasoning modules to model the problem constraints (prior knowledge), and the components of the objective function. De facto, these examples illustrate how to develop "gadgets" to encode a variety of constraints and prior knowledge in DRNets. **(3)** We demonstrate the potential of DRNets on two de-mixing tasks with detailed experimental results. We show how **(3.1)** DRNets significantly outperformed the state of the art and human experts on **Crystal-Structure-Phase-Mapping instances**, recovering more precise, interpretable, and physically meaningful crystal structure pattern decompositions. In this task, DRNets solve a previously *unsolved chemical system*, which subsequently led to the discovery of a new material that is important

for solar fuels technology. **(3.2)** On **Multi-MNIST-Sudoku instances**, without direct supervision, DRNets perfectly recovered the digits in the mixed Sudokus with 100% digit accuracy, outperforming the *supervised* state-of-the-art MNIST de-mixing models, including CapsuleNet (Sabour et al., 2017) and ResNet (He et al., 2016).

## 2 RELATED WORK

DRNets have been motivated by scientific tasks such as crystal phase mapping that involve identifying or de-mixing patterns in data that satisfy prior scientific knowledge. In general, for such tasks there are no labeled datasets. So our work focus on **unsupervised or weakly supervised learning, using prior knowledge.**

**Most closely related work: Unsupervised or weakly supervised de-mixing approaches.** Pattern de-mixing approaches have been developed under the name of *source separation* in the signal processing community. The unsupervised methods in this area mostly try to solve the de-mixing, which is in general ill-posed, using different regularizations. Among existing methods, recent work for weakly supervised audio source separation (Zhang et al., 2017) is most related to DRNets since they also employed a generative adversarial network (GAN) in their model. However, their model mainly employs the decoder of GAN to discriminate the reality of separated sources, while DRNets only utilize the generator of GAN as the generative model of possible sources. Moreover, the weakly supervised setting in their paper is actually too strong: they need the true labels of mixed sources, which is almost the goal of our tasks, and therefore it is not applicable to our settings. We now consider the state-of-the-art models for the tasks considered in this paper. For **Crystal-structure-phase-mapping**, due to the lack of labeled datasets, existing models (Ermon et al., 2015; Xue et al., 2017; Bai et al., 2017; 2018; Stanev et al., 2018) are mainly based on non-negative matrix factorization (NMF), which is in general unsupervised. Stanev et al. (2018) proposed the NMF-k algorithm, which applies a customized clustering process over the results of thousands of runs of pure NMF algorithm (Long et al., 2009) to cluster the common phase patterns. However, NMF-k does not enforce prior knowledge (namely thermodynamic rules) and therefore the solutions produced are often not completely physically meaningful. To address this limitation several approaches have been developed that use external mixed-integer programming modules to interact with the NMF de-mixing module to enforce prior knowledge (Ermon et al., 2015; Bai et al., 2017; 2018). However, the coordination barrier between the NMF de-mixing module and the reasoning module often results in inferiror performance, where the solution satisfies constraints at the cost of huge reconstruction loss. In contrast to existing models, DRNets seamlessly integrate the pattern de-mixing module and the reasoning module, recovering almost exact ground truth decomposition. In our experiments we thoroughly compare DRNets' performance against the state of the art (IAFD and NMF-k) for crystal-structure pattern de-mixing. **MNIST de-mixing** was first studied by Hinton et al. in 2000, where the aim is to identify or de-mix overlapping digits coming from the MNIST datasets (LeCun et al., 1998). More recently, it has been tackled with state-of-the-art neural network models such as CapsuleNet (Sabour et al., 2017) and ResNet (He et al., 2016). Existing works concerning this task are mainly in supervised settings, where we have labels of digits for each overlapping image. However, in this paper, we aim to tackle this task in a weakly supervised setting, where we only have access to the prototypes of single digits and the extra Sudoku rules. Due to the lack of existing models with the same setting, we compared DRNets's performance against the state-of-the-art supervised models (CapsuleNet and ResNet). By utilizing the supervision from prior knowledge and reasoning, we show that DRNets' outperformed all supervised models with 100% digit accuracy.

**Enhancing deep learning with symbolic prior knowledge.** Exploiting problem structure and reasoning about prior knowledge has been of increasing interest to facilitate deep learning (Garcez et al., 2019). In computer vision, symmetry constraints, bone-length constraints and linear constraints were introduced for human pose estimation (Zhou et al., 2017; 2016) and image segmentation (Pathak et al., 2015) to regularize the output and enhance generalization. In natural language processing, Hu et al. (2016a;b) introduced the *posterior regularization* (Ganchev et al., 2010) framework into deep learning to incorporate rule-based grammatical knowledge using first order logic. Xu et al. (2017) proposed a semantic loss function to enforce propositional logic constraints on the output of neural networks for semi-supervised multi-class classification tasks. Wang et al. (2019) proposed SATNet, which approximately encodes a MAXSAT solver into a neural network layer called SATNet layer, to explicitly learn the logical structures (e.g., parity function and Sudoku) from the labeled training data.

Previous works in this area primarily focus on supervised or semi-supervised settings for data-rich domains, where direct supervision from labels reduce the importance of explicitly reasoning about prior knowledge. In contrast, with an unsupervised setting, the supervision of DRNets comes from reasoning about prior knowledge and self-reconstruction, which is strongly desired for problems in scientific discovery due to the lack of labeled datasets, and strongly motivated by extensive prior knowledge from sources ranging from fundamental principles to the intuitive experience of scientists.

Among existing works, SATNet is mostly related to DRNets in the sense of bridging logical reasoning with deep learning. However, SATNet is essentially designed for learning logical structures (prior knowledge) from labeled training examples while DRNets aim to facilitate unsupervised learning with known logical constraints. In terms of the encoding of the reasoning module, the semantic loss (Xu et al., 2017) is mostly related to ours. However, the semantic loss encodes constraints by propositional logic, which requires enumerating all possible Boolean assignments that satisfy the constraints. Consequently, the semantic loss has to enumerate a large number of assignments to encode constraints such as k-sparsity constraints and All-Different constraints, which is not applicable to tasks considered in this paper.

## 3 DEEP REASONING NETWORKS

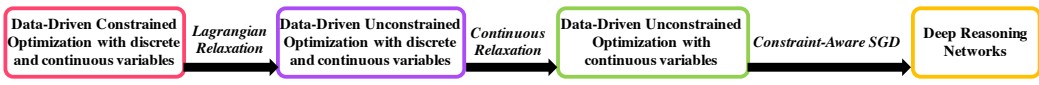

Figure 3: The reduction flow of Deep Reasoning Networks.

DRNets (see Fig.2) are inspired by human thinking (Shivhare & Kumar, 2016): we abstract patterns to higher-level descriptions and combine them with prior-knowledge to fill-in the gaps. Consider the Multi-MNIST-Sudoku example (Fig.1): we first guess the digits in each cell based on the patterns; we re-adjust our initial beliefs and re-image the overlapping patterns by reasoning about Sudoku rules and comparing them to the original ones, potentially involving several iterations. Analogously, in a reasoning system, an inference procedure derives what follows from an initial set of axioms and rules. For example, in a standard 9x9 Sudoku, an inference procedure identifies the missing cell values of the input Sudoku. A constraint solver is a particular type of reasoning system in which axioms and rules are expressed as constraints and the inference procedure is a search method.

Formally, **DRNets formulate unsupervised pattern de-mixing as a data-driven constrained optimization, incorporating abstractions and reasoning about structure and prior knowledge:**

$$\min_{\theta} \ \frac{1}{N}\sum_{i=1}^{N}\mathcal{L}(G(\phi_\theta(\mathbf{x}_i)),\mathbf{x}_i) \quad \text{s.t. } \phi_\theta(\mathbf{x}_i)\in\Omega^{\text{local}} \text{ and } (\phi_\theta(\mathbf{x}_1),...,\phi_\theta(\mathbf{x}_N))\in\Omega^{\text{global}} \tag{1}$$

In this formulation, $\mathbf{x}_i \in R^n$ is the $i$-th $n$-dimensional input data point, $\phi_\theta(\cdot)$ is the function of the encoder in DRNets parameterized by $\theta$, $G(\cdot)$ denotes the generative decoder, $\mathcal{L}(\cdot,\cdot)$ is the loss function (e.g., evaluating the reconstruction of patterns), $\Omega^{\text{local}}$ and $\Omega^{\text{global}}$ are the constrained spaces w.r.t. a single input data point and several input data points, respectively. $G(\cdot)$ is in general a fixed pre-trained or parametric model. For example, in Multi-MNIST-Sudoku, $G(\cdot)$ is a pre-trained conditional GAN (Mirza & Osindero, 2014) using hand-written digits, and for Crystal-Structure-Phase-Mapping, $G(\cdot)$ is a Gaussian Mixture model. Note that constraints can involve several (potentially all) data points: e.g., in Sudoku, all digits should form a valid Sudoku and in crystal-structure-phase-mapping, all data points in a composition graph should form a valid phase diagram. Thus, we specify local and global constraints in DRNets – local constraints only involve a single input data point whereas global constraints involve several input data points, and they are optimized using different strategies.

Solving the constrained optimization problem (1) directly is extremely challenging since the objective function in general involves deep neural networks, which are highly non-linear and non-convex, and prior knowledge often even involves combinatorial constraints (Fig.3). Therefore, we use Lagrangian relaxation to approximate equation (1) with an unconstrained optimization problem, i.e.,

$$\min_{\theta} \ \frac{1}{N}\sum_{i=1}^{N}\mathcal{L}(G(\phi_\theta(\mathbf{x}_i)),\mathbf{x}_i) + \lambda^l\psi^l(\phi_\theta(\mathbf{x}_i)) + \sum_{j=1}^{N_g}\lambda_j^g\psi_j^g(\{\phi_\theta(\mathbf{x}_k)|k\in S_j\}) \tag{2}$$

$N$ is the number of input data points, $N_g$ denotes the number of global constraints, $S_j$ denotes the set of indices w.r.t. the data points involved in the $j$-th global constraint, and $\psi^l, \psi_j^g$ denote the penalty

functions for local constraints and global constraints, respectively, along with their corresponding penalty weights $\lambda^l$ and $\lambda_j^g$. In the following, we propose two mechanisms to tackle the above unconstrained optimization task (Fig.3).

**Continuous Relaxation:** Prior knowledge often involves combinatorial constraints with discrete variables that are difficult to optimize in an end-to-end manner using gradient-based methods. Therefore, we need to design proper continuous relaxations for discrete constraints to make the overall objective function differentiable. Existing works (Hu et al., 2016a; Xu et al., 2017) proposed several relaxations for injecting first-order logic and propositional logic into deep learning. However, limited by the expressive power of those logic formulas, we need a large number of logical terms to express constraints such as k-sparsity constraints or All-Different constraints. Therefore, to instantiate DRNets for our tasks, we propose a group of entropy-based continuous relaxations to encode general discrete constraints such as sparsity, cardinality and All-Different constraints (see Fig.4). We construct continuous relaxations based on probabilistic modelling of discrete variables,

| Cardinality Constraint | Cardinality Constraint Relaxation |
|---|---|
| $e_{i,j} \in \{0,1\}$  $j = 1 \dots 8$ | $\min_\theta H(P_i) + H(Q_i)$ |
| $s.t. \sum_{j=1}^4 e_{i,j} = 1$ and $\sum_{j=5}^8 e_{i,j} = 1$ | $= -\sum_{j=1}^4 P_{i,j} \log P_{i,j} - \sum_{j=1}^4 Q_{i,j} \log Q_{i,j}$ |
| **All-Different Constraint** | **All-Different Constraint Relaxation** |
| For all constrained set $S$ | For all constrained set $S$ |
| $s.t. \sum_{i \in S} e_{i,j} = 1$ for j = 1 ... 8 | $\max_\theta H(\bar{P}_S) + H(\bar{Q}_S)$ |
| $k$**-Sparsity Constraint** | $k$**-Sparsity Constraint Relaxation** |
| $e_{i,j} \in \{0,1\}$  $j = 1 \dots M$  $s.t. \sum_{j=1}^M e_{i,j} \le k$ | $\min_\theta max\{H(P_M), c\}$,  where $c < \log k$ |

Figure 4: Examples of continuous relaxations: $e_{i,j}, P_i, Q_i, P_M$ denote binary variables, the discrete distribution over digits 1 to 4, the discrete distribution over digits 5 to 8, and the discrete distribution over values 1 to $M$.

where we model a probability distribution over all possible values for each discrete variable. For example, in Multi-MNIST-Sudoku, a way of encoding the possible two digits in the cell indicated by data point $x_i$ (one from $\{1...4\}$ and the other from $\{5...8\}$), is to use 8 binary variables $e_{i,j} \in \{0,1\}$, while requiring $\sum_{j=1}^4 e_{i,j} = 1$ and $\sum_{j=5}^8 e_{i,j} = 1$. In DRNets, we model probability distribution $P_i$ and $Q_i$ over digits 1 to 4 and 5 to 8 respectively: $P_{i,j,j=1...4}$ and $Q_{i,j,j=1...4}$ denote the probability of digit $j$ and the probability of digit $j + 4$, respectively. We approximate the cardinality constraint of $e_{i,j}$ by minimizing the entropy of $P_i$ and $Q_i$, which encourages $P_i$ and $Q_i$ to collapse to one value. Another combinatorial constraint in Multi-MNIST-Sudoku is the All-Different constraint, where all the cells in a *constrained set* $S$, i.e., each row, column, and any of four 2x2 boxes involving the corner cells, must be filled with non-repeating digits. For a probabilistic relaxation of the All-Different constraint, we analogously define the entropy of the averaged digit distribution for all cells in a constrained set $S$, i.e., $H(\bar{P}_S)$ :

$$H(\bar{P}_S) = -\sum_{j=1}^4 \bar{P}_{S,j} \log \bar{P}_{S,j} = -\sum_{j=1}^4 \left( \frac{1}{|S|} \sum_{i \in S} P_{i,j} \right) \log \left( \frac{1}{|S|} \sum_{i \in S} P_{i,j} \right) \tag{3}$$

In this equation, a larger value implies that the digits in the cells of $S$ distribute more uniformly. Thus, we can analogously approximate All-Different constraints by maximizing $H(\bar{P}_S)$ and $H(\bar{Q}_S)$. One can see, by minimizing all $H(P_i)$ and $H(Q_i)$ to 0 as well as maximizing all $H(\bar{P}_S)$ and $H(\bar{Q}_S)$ to $\log |S|$, we find a valid solution for the two 4x4 Sudoku puzzles, where all $P_{i,j}$ are either 0 or 1.

We also relax $k$-sparsity constraints, which for example in Crystal-Phase-Mapping state the maximum number $k$ of pure phases in an XRD-pattern, by minimizing the entropy of the phase distribution $P_M$ below a threshold $c < \log k$. We choose the threshold $c < \log k$ because the entropy of a discrete distribution $P_M$ concentrated on at most $k$ values cannot exceed $\log k$. Note that other relaxations can be adapted in DRNets, for these and other tasks. See also additional relaxations (e.g., for SAT constraints), detailed relaxation derivations, and implementation details in supplementary materials.

**Constraint-Aware Stochastic Gradient Descent:** We introduce a variant of standard SGD method called constraint-aware SGD, which is conceptually similar to the optimization process in GraphRNN (You et al., 2018), to tackle the optimization of global penalty functions $\psi_j^g(\{\phi_\theta(\mathbf{x}_k) | k \in S_j\})$, which involve several (potentially all) data points. We define a *constraint graph*, an undirected

---

**Algorithm 1** Constraint-aware stochastic gradient descent optimization of deep reasoning networks.

---

**Input:** **(i)** Data points $\{x_i\}_{i=1}^N$. **(ii)** Constraint graph. **(iii)** Penalty functions $\psi^l(\cdot)$ and $\psi_j^g(\cdot)$ for the local and the global constraints. **(iv)** Pre-trained or parametric generative decoder $G(\cdot)$.

1: Initialize the penalty weights $\lambda^l, \lambda_j^g$ and thresholds for all constraints.
2: **for** number of optimization iterations **do**
3:  Batch data points $\{\mathbf{x}_1, ..., \mathbf{x}_m\}$ from the sampled (maximal) connected components.
4:  Collect the global penalty functions $\{\psi_j^g(\cdot)\}_{j=1}^M$ concerning those data points.
5:  Compute the latent space $\{\phi_\theta(\mathbf{x}_1), ..., \phi_\theta(\mathbf{x}_m)\}$ from the encoder.
6:  Adjust the penalty weights $\lambda_l, \lambda_j^g$ and thresholds accordingly.
7:  minimize $\frac{1}{m}\big(\sum_{i=1}^m \mathcal{L}(G(\phi_\theta(\mathbf{x}_i)), \mathbf{x}_i) + \lambda_l \psi^l(\phi_\theta(\mathbf{x}_i))\big) + \sum_{j=1}^M \lambda_j^g \psi_j^g(\{\phi_\theta(\mathbf{x}_k)|k \in S_j\})$
   using any standard gradient-based optimization method and update the parameters $\theta$.
8: **end for**

---

graph in which each data point forms a vertex and two data points are linked if they are in the same global constraint. Constraint-aware SGD batches data points from the randomly sampled (maximal) connected components in the *constraint graph*, and optimizes the objective function w.r.t. the subset of global constraints concerning those data points and the associated local constraints. For example, in Multi-MNIST-Sudoku, each overlapping Sudoku forms a maximal connected component, we batch the data points from several randomly sampled overlapping Sudokus and optimize the All-Different constraints (global) as well as the cardinality constraints (local) within them. However, in Crystal-Structure-Phase-Mapping, the maximal connected component becomes too large to batch together, due to the constraints (*phase field connectivity* and *Gibbs-alloying rule*) concerning all data points in the composition graph. Thus, we instead only batch a subset (still a connected component) of the maximal connected component – e.g., a path in the composition graph, and optimize the objective function that only concerns constraints within the subset (along the path). By iteratively solving sampled local structures of the "large" maximal component, we cost-efficiently approximate the entire global constraint. Moreover, for optimizing the overall objective, constraint-aware SGD dynamically adjusts the thresholds and the weights of constraints according to their satisfiability, which can involve non-differentiable functions (See details in appendix). For efficiency and potential capability of generalization, DRNets solve all instances together using constraint-aware SGD (see Algorithm 2).

## 4 EXPERIMENTS

We illustrate the power of DRNets mainly on two pattern de-mixing tasks – disentangling two overlapping hand-written Sudokus (**Multi-MNIST-Sudoku**) and inferring crystal structures of materials from X-ray diffraction data (**Crystal-Structure-Phase-Mapping**). Limited by the space, we put the details of the experiments and the experimental results of DRNets on other tasks in supplementary material. Note that, since DRNets are an unsupervised framework, we can apply the *restart* (Gomes et al., 1998) mechanism, i.e., we can re-run DRNets for unsolved instances.

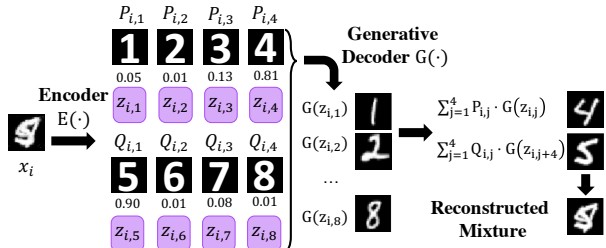

| Method | Accuracy (%) | | Time |
|---|---|---|---|
| | Digit | Sudoku | |
| DRNets (Optimization w/ Restart) | **100.0** | **100.0** | 50min |
| DRNets (Optimization) | **99.9** | 98.6 | 28min |
| DRNets (Optimization w/o Reasoning) | 88.8 | 15.0 | 110min |
| DRNets (Generalization) | 98.0 | 75.7 | 13min+4hrs |
| CapsuleNet | 97.9 | 50.9 | 1min+30min |
| CapsuleNet + local search | 97.9 | 57.8 | 3hrs+30mins |
| ResNet-18 | 97.7 | 68.5 | 3min+10hrs |
| ResNet-18 + local search | 97.7 | 88.3 | 3hrs+10hrs |

Figure 5: Left: The latent space of DRNets for Multi-MNIST-Sudoku. Right: Accuracy comparison. We show "test time + training time" for supervised baselines and the generalization mode of DRNet, and "solving time" for the optimization mode of DRNets. (See also supplementary materials.)

**Multi-MNIST-Sudoku:** We generated 160,000 input data points for each training set, validation set and test set, where each data point corresponds to a 32x32 image of overlapping digits coming from MNIST (LeCun et al., 1998) and every 16 data points form a 4-by-4 overlapping Sudokus. For Multi-MNIST-Sudoku, DRNets batch every 16 data points together to enforce the All-Different constraints among the cells of each Sudoku. The encoder of DRNets is composed of two ResNet-18 He et al. (2016) and we use a conditional GAN (Mirza & Osindero, 2014) as our generative decoder (denoted as $G(\cdot)$), which is trained using the digits in the *training set* of MNIST. For each cell $\mathbf{x}_i$, the encoder encodes a latent space, which consists of two parts: The first part includes two distribution $P_i$ and $Q_i$ (see Fig.5) concerning the possible digits in the cell, and the second part is the latent encodings $z_{i,1}, ..., z_{i,8}$ of each possible digit conditioned on the overlapping digits, which is used by the generative decoder to generate the corresponding digits $G(z_{i,j})$. We estimate the two digits in the cell by computing the expected digits over $P_i$ and $Q_i$, i.e., $\sum_{j=1}^{4} P_{i,j}G(z_{i,j})$ and $\sum_{j=1}^{4} Q_{i,j}G(z_{i,j+4})$, and reconstruct the original input mixture (see Fig.5). As described above, we impose the continuous relaxation of the cardinality and All-Different constraints to reason about the the Sudoku structure among cells of the overlapping Sudokus. To demonstrate the power of reasoning, we compared our unsupervised DRNets with supervised start-of-the-art MNIST de-mixing models – CapsuleNet (Sabour et al., 2017) and ResNet (He et al., 2016), and a variant of DRNets that removes the reasoning modules ("DRNets w/o Reasoning"). To saturate the performance of baseline models, we also applied a post-process local search for them to incorporate the Sudoku Rules. Specifically, we did a local search for the top-2 (top-3 would take too long to search) most likely choice of digits for each Sudoku of the two overlapping Sudokus and try to satisfy Sudoku rules with minimal modification compared with the original prediction. We evaluate both the percentage of digits that are correctly de-mixed (digit accuracy) and the percentage of overlapping Sudokus that have all digits correctly de-mixed (Sudoku accuracy). Empowered by reasoning, DRNets significantly outperformed CapsuleNet, ResNet, and DRNets without reasoning, perfectly recovered all digits with the *restart* mechanism (see Fig.5), and additionally reconstructed the mixture with high-quality (see Fig.1). Moreover, because DRNets solve all instances together (see Algorithm 2), not only can DRNets solve instances directly on the test set from random initialization, DRNets can also generalize from the training set to test set, given enough training examples. DRNets learn to generalize its de-mixing performance on the test set by solving the training set instances *self-supervised* (Jing & Tian, 2019) by Sudoku rules, instead of labels, and even outperform CapsuleNet and ResNet (Fig.5). Note that, for unseen instances in the test set, we further optimize the instances for 25 steps to achieve the reported performance (Additional details in the supplementary material).

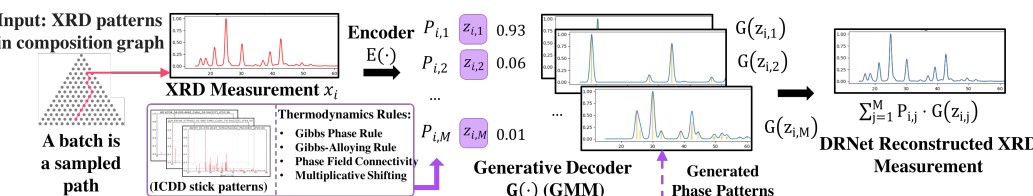

Figure 6: The latent space of DRNets for Crystal-Structure-Phase-Mapping. $M$ denotes the number of possible phases. (For Al-Li-Fe, $M = 159$; For Bi-Cu-V, $M = 100$.)

**Crystal-Structure-Phase-Mapping** concerns inferring crystal structures from a set of X-ray diffraction measurements (XRDs) of a given chemical system, satisfying thermodynamic constraints. Crystal structure phase mapping is a very challenging task, a major bottleneck in high-throughput materials discovery: Each X-ray measurement may involve several mixed crystal structures; each chemical system includes hundreds of possible crystal structures; for each crystal structure pattern, we only have a theoretical (idealized) model of pure crystal phases; the thermodynamic rules are also complex; and the crystal patterns are difficult for human experts to interpret. Herein, we illustrate DRNet for crystal structure phase mapping for two chemical systems: (1) a ternary **Al-Li-Fe** oxide system (Le Bras et al., 2014), which is theoretically based, synthetically generated, with ground truth solutions, and (2) a ternary **Bi-Cu-V** oxide system, which is a more challenging real experiment-based system, more noisy and uncertain. For each system, each input data point is the XRD of a mixture of crystal structures. Additionally, the input includes the *composition graph* specifying elemental compositions and the *constraint graph* of the data points. We also collected a library of possible crystal structures from the International Centre for Diffraction Data (ICDD) database. Each crystal

structure (also named *phase*) is given as a list of diffraction peak location-amplitude pairs, (referred to as *stick pattern*), representing the ideal phase patterns measured in a perfect condition (see Fig.6). To model more realistic conditions, DRNets simulate the real phase patterns from *stick patterns* using Gaussian mixture models, where the relative peak locations and mixture coefficients are given by the stick locations and amplitudes. Moreover, the peak width, peak location shift, and peak amplitude variance are parameterized by the latent encoding $z_{i,j}$ and used by the generative decoder to generate the corresponding possible phase patterns in the reconstructed XRD measurement.

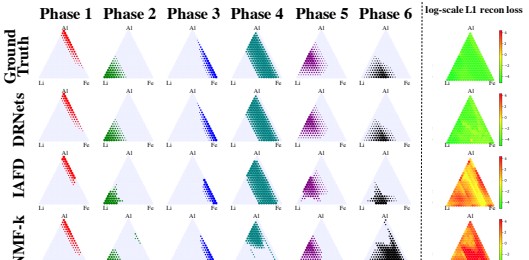

| Chemical Systems: | Reconstruction Losses | | Phase Fidelity Loss | Thermodynamic Rules Satisfaction (Percentage of data points / phase field that satisfy each constraint) | | |
|---|---|---|---|---|---|---|
| **Al-Li-Fe** | L1 Loss | L2 Loss | JS distance ($\times 10^{-2}$) | Gibbs | Gibbs-Alloy | Phase Field Connectivity |
| DRNets | **0.039** | **<0.001** | **<0.001** | 100% | 100% | 100% |
| IAFD | 5.994 | 0.535 | 11.30 | 100% | 100% | 100% |
| NMF-k | 7.267 | 0.438 | 56.10 | 94% | 87% | 71% |
| **Bi-Cu-V** | L1 Loss | L2 Loss | JS distance ($\times 10^{-2}$) | Gibbs | Gibbs-Alloy | Phase Field Connectivity |
| DRNets | **3.993** | **0.196** | **8.370** | 100% | 100% | 100% |
| IAFD | 7.425 | 0.545 | 93.36 | 100% | 99% | 95% |
| NMF-k | 8.033 | 0.675 | 92.63 | 51% | 35% | 83% |

Figure 7: Left: Comparison of phase concentration and reconstruction loss for different methods in Al-Li-Fe oxide system. Note that, 6 pure phases (out of 159 possible candidates) appear in the system and result in 15 different mixtures. Each dot represents an XRD measurement whose size is proportional to the estimated phase concentration. DRNet's phase concentration closely match the ground truth in contrast to IAFD's and NMF-k's. The heatmap on the right shows that DRNets reconstruct the XRD measurements much better than other methods with respect to the L1 loss. Right: DRNets outperform both IAFD and NMF-k with better reconstruction error and perfect rule satisfaction on both systems. (additional details for Bi-Cu-V in the supplementary material).

We compared DRNets with IAFD (Bai et al., 2017) and NMF-k (Stanev et al., 2018), which are both state-of-the-art non-negative matrix factorization (NMF) based unsupervised de-mixing models. NMF-k improves the pure NMF algorithm (Long et al., 2009) by clustering common phase patterns from thousands of runs. However, NMF-k does not directly enforce thermodynamic rules and therefore the solutions produced are often not completely physically meaningful. IAFD uses external mixed-integer programming modules to enforce thermodynamic rules during the de-mixing. However, due to the gap between the external optimizer and NMF module, the solution of IAFD is still far from the ground truth. Our evaluation criteria include reconstruction losses, phase fidelity loss and the satisfaction of thermodynamic rules. Note that, the phase fidelity loss measures the JS distance between the de-mixed phases and the closest ideal phases by fitting the de-mixed phases with the ICDD stick patterns using the physical model (Le Bras et al., 2014). As shown in Fig.7, for the Al-Li-Fe oxide system, the phase concentration (the distribution of de-mixed pure phases over all data points of that chemical system) of either IAFD or NMF-k is far from the ground truth. In contrast, DRNet almost exactly recovered the ground truth solution by seamlessly integrating pattern recognition, reasoning and prior knowledge. Moreover, by explicitly incorporating the ICDD stick pattern information into DRNets, the phases de-mixed by DRNets are much more real than those from IAFD and NMF-k (see phase fidelity loss). For Bi-Cu-V oxide system, DRNets solved this *previously unsolved* real system, producing valid crystal structures and significantly outperforming IAFD and NMF-k w.r.t. reconstruction errors and phase fidelity loss. In addition, materials science experts thoroughly checked DRNet's solution of Bi-Cu-V oxide system, approved it, and subsequently discovered a new material that is important for solar fuels technology.

## 5 CONCLUSIONS AND FUTURE WORK

We propose DRNets, a powerful end-to-end framework that combines deep learning with logical and constraint reasoning for solving unsupervised pattern de-mixing tasks. DRNets outperform the state of the art for de-mixing MNIST Sudokus and crystal-structure phase mapping, solving previously unsolved chemical systems substantially beyond the reach of other methods and materials science experts' capabilities. While we illustrate the potential of DRNets with unsupervised settings, it is straightforward to impose supervision into DRNets. Future research includes exploring DRNets

for incorporating other types of constraints, prior knowledge, and objective functions, for other applications.

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

# A SUPPLEMENTARY MATERIALS

Herein, we provide additional details about DRNets and our experimental settings for a better understanding of DRNets and reproducibility of our results. Code and datasets to reproduce the experiments will be provided with the final version of the paper.

## A.1 CONTINUOUS RELAXATION

In this section, we provide more relxations for other constraints such as SAT constraints and provide an intuitive high-level informal proof that all the relaxations converge to a valid solution of the discrete version when it achieves its minimal value. Fig.8 summarizes the relaxations.

| Cardinality Constraint | Cardinality Constraint Relaxation |
|---|---|
| $e_{i,j} \in \{0,1\}$   $j = 1 \dots 8$ | $\min_{\theta} H(P_i) + H(Q_i)$ |
| $s.t. \sum_{j=1}^{4} e_{i,j} = 1$ and $\sum_{j=5}^{8} e_{i,j} = 1$ | $= -\sum_{j=1}^{4} P_{i,j} \log P_{i,j} - \sum_{j=1}^{4} Q_{i,j} \log Q_{i,j}$ |
| **All-Different Constraint** | **All-Different Constraint Relaxation** |
| For all constrained set $S$ | For all constrained set $S$ |
| $s.t. \sum_{i \in S} e_{i,j} = 1$ for $j = 1 \dots 8$ | $\max_{\theta} H(\bar{P}_S) + H(\bar{Q}_S)$ |
| **$k$-Sparsity Constraint** | **$k$-Sparsity Constraint Relaxation** |
| $e_{i,j} \in \{0,1\}$   $j = 1 \dots M$   $s.t. \sum_{j=1}^{M} e_{i,j} \leq k$ | $\min_{\theta} max\{H(P_M), c\}$,  where $c < \log k$ |
| **Integer Programming Encoding of SAT** | **SAT Relaxation** |
| For any literal $x_i$ and its negation $\bar{x}_i$, | For any literal $x_i$ and its negation $\bar{x}_i$ ($i = 1 \dots N_l$), we model |
| $s.t. x_i, \bar{x}_i \in \{0,1\}$ and $x_i + \bar{x}_i = 1$ | a distribution $B_i \sim \text{Bern}(p_i, q_i)$, $s.t.$  $x_i \triangleq p_i$, and $\bar{x}_i \triangleq q_i$ |
| For any clause $C_j = a_{j,1} \vee \cdots \vee a_{j,K_j}$ | For all clause $C_j = a_{j,1} \vee \cdots \vee a_{j,K_j}$, $j = 1 \dots N_c$ |
| $\forall a_{j,k} \in \{x_i, \bar{x}_i\}_{i=1}^{n}$,   $s.t. \sum_{k=1}^{K_j} a_{j,k} \geq 1$ | $\min_{\theta} \sum_{j=1}^{N_c} leaky\_relu \left(1 - \sum_{j=1}^{K_j} a_{j,k}\right) + \lambda_{\text{h}} \sum_{i=1}^{N_l} H(B_i)$ |

Figure 8: Examples of continuous relaxations: $e_{i,j}, P_i, Q_i, P_M, N_c, N_l, K_j, \lambda_h, B_i$ denote binary variables, the discrete distribution over digits 1 to 4, the discrete distribution over digits 5 to 8, the discrete distribution over values 1 to $M$, the number of clauses, the number of literals, the number of literals in the $j$-th clause, the weights of entropy terms, and the Bernoulli distribution for the $i$-th literal. "leaky_relu" is the leaky ReLU.

For cardinality constraints, when the entropy of distribution $P_i$ and $Q_i$ reaches 0, all the probability mass collapses to only one variable. Therefore, all $P_{i,j}$ and $Q_{i,j}$ are either 0 or 1, which is a valid solution of the original discrete constraints.

For All-Different constraints, we maximize the entropy of the averaged digit distribution for all cells in a constrained set $S$, i.e., $H(\bar{P}_S)$. Note that, the All-Different constraints are imposed together with the cardinality constraints. Therefore, when the entropy of the digit distribution in each cell is zero, we know that the digit distribution of each cell converges to one digit. Hence, if $H(\bar{P}_S)$ reaches its maximum, i.e., $\log |S|$, we have $\frac{1}{|S|} \sum_{i \in S} P_{i,j} = \frac{1}{|S|}$ for all digit $j$. Crossed with the fact that $P_{i,j}$ are either 0 or 1 when the cardinality constraints are satisfied, we know that only one $P_{i,j}$ is equal to 1 for all cell $i$ in the set $S$ and others are 0, which directly states the All-Different constraints.

We derive the k-Sparsity constraints in a similar way as the cardinality constraints except that we now want to force the distribution to concentrate on at most k digits. By normalizing the values of

discrete variables $e_{i,j}$ $(j = 1...M)$ to a discrete distribution $P_M$, we can minimize the entropy of distribution $P_M$ to at most $\log k$, which is the maximal entropy when the distribution concentrates on only $k$ values. Though, $H(P_M) < \log k$ is not a sufficient condition for k-sparsity, we can initialize the threshold $c$ of k-sparsity constraints to $\log k$ and dynamically adjust the value of $c$ based on the satisfaction of the k-sparsity constraints. In practice, it works well with the supervision from other modules, such as the self-reconstruction.

For SAT constraint relaxations, the key idea is to minimize the entropy of the Bernoulli distribution over each literal to force it converge to either 1 or 0. Then, we maximize the sum of the value of literals in each clause (or their negation) to encourage one of the literals to be 1. However, maximizing the sum of the value of literals does not necessarily give you a valid assignment because there could exist an assignment that the sum of literals in some clauses are 0 and the sum of literals in other clauses are very large. Therefore, we use leaky_rule (Xu et al., 2015) function to discount the loss when the sum is larger than 1. As shown in Fig.8, we formulate the relaxation loss function in a form to be minimized. For k-SAT problems with $N_c$ clauses, we can set the leaky ratio to be $\frac{1}{N_c k}$, so that any invalid assignment cannot have a loss that is less or equal to 0. On the other hand, for any valid assignment, the sum of literals in each clause is at least 1. Thus, we can obtain a valid assignment of k-SAT constraints by minimizing the loss function to 0.

We describe other task specific constraints (e.g., phase field connectivity constraints) in the following experimental sections.

## A.2 CONSTRAINT-AWARE STOCHASTIC GRADIENT DESCENT:

---

**Algorithm 2** Constraint-aware stochastic gradient descent optimization of deep reasoning networks.

---

**Input:** (i) Data points $\{x_i\}_{i=1}^N$. (ii) Constraint graph. (iii) Penalty functions $\psi^l(\cdot)$ and $\psi_j^g(\cdot)$ for the local and the global constraints. (iv) Pre-trained or parametric generative decoder $G(\cdot)$.

1: Initialize the penalty weights $\lambda^l, \lambda_j^g$ and thresholds for all constraints.
2: **for** number of optimization iterations **do**
3:     Batch data points $\{\mathbf{x}_1, ..., \mathbf{x}_m\}$ from the sampled (maximal) connected components.
4:     Collect the global penalty functions $\{\psi_j^g(\cdot)\}_{j=1}^M$ concerning those data points.
5:     Compute the latent space $\{\phi_\theta(\mathbf{x}_1), ..., \phi_\theta(\mathbf{x}_m)\}$ from the encoder.
6:     Adjust the penalty weights $\lambda_l, \lambda_j^g$ and thresholds accordingly.
7:     minimize $\frac{1}{m}\left(\sum_{i=1}^m \mathcal{L}(G(\phi_\theta(\mathbf{x}_i)), \mathbf{x}_i) + \lambda_l \psi^l(\phi_\theta(\mathbf{x}_i))\right) + \sum_{j=1}^M \lambda_j^g \psi_j^g(\{\phi_\theta(\mathbf{x}_k)|k \in S_j\})$
    using any standard gradient-based optimization method and update the parameters $\theta$.
8: **end for**

---

We introduce a variant of standard SGD method called constraint-aware SGD, which is conceptually similar to the optimization process in GraphRNN (You et al., 2018), to tackle the optimization of global penalty functions $\psi_j^g(\{\phi_\theta(\mathbf{x}_k)|k \in S_j\})$, which involve several (potentially all) data points. We define a *constraint graph*, an undirected graph in which each data point forms a vertex and two data points are linked if they are in the same global constraint. Constraint-aware SGD batches data points from the randomly sampled (maximal) connected components in the *constraint graph*, and optimizes the objective function w.r.t. the subset of global constraints concerning those data points and the associated local constraints. For example, in Multi-MNIST-Sudoku, each overlapping Sudoku forms a maximal connected component, we batch the data points from several randomly sampled overlapping Sudokus and optimize the All-Different constraints (global) as well as the cardinality constraints (local) within them. However, in Crystal-Structure-Phase-Mapping, the maximal connected component becomes too large to batch together, due to the constraints (*phase field connectivity* and *Gibbs-alloying rule*) concerning all data points in the composition graph. Thus, we instead only batch a subset (still a connected component) of the maximal connected component – e.g., a path in the composition graph, and optimize the objective function that only concerns constraints within the subset (along the path). By iteratively solving sampled local structures of the "large" maximal component, we cost-efficiently approximate the entire global constraint.

Moreover, for optimizing the overall objective, constraint-aware SGD dynamically adjusts the thresholds and the weights of constraints according to their satisfiability, which can involve non-differentiable functions. Specifically, we initialize penalty weights of constraints and thresholds for

penalty functions using hyper-parameters. During training, we check the satisfiability of constraints (this step could involve non-differentiable functions) after several epochs and increase the penalty for violated constraints. For example, the threshold $c$ of k-sparsity is initialized as $\log k$, which is the entropy of the case that the probability mass is evenly distributed among $k$ entities. Thus, it could be the case that there are more than $k$ entities, but their probability mass is not evenly distributed. Hence, we check the satisfiability of k-sparsity constraint: if the entropy is already below the current threshold ($\log k$) and there are still more than $k$ entities with probability mass more than $\epsilon$ (0.01), we decrease the threshold $c$ to keep enforcing the model to minimize the entropy to reach the k-sparsity.

Finally, to better exploit parallelization, DRNets solve all instances together using constraint-aware SGD (see Algorithm 2).

### A.3 RESTART MECHANISM FOR DRNETS:

Note that, since DRNets are an unsupervised framework, we can apply the *restart* (Gomes et al., 1998) mechanism, i.e., we can re-run DRNets for unsolved instances. Specifically, since DRNets directly incorporate logical constraints, we can check whether those constraints are satisfied at the end of a run. If not, for instances with violated constraints, we re-run the algorithm again on them. We only applied *restart* mechanism on Multi-MNIST-Sudoku and other NP-C problems (in the appendix) such as 3-SAT problems and standard Sudoku problems. For crystal-structure phase mapping, the results generated from one run of DRNets is already good enough.

### A.4 EXPERIMENTAL CONFIGURATION

All the experiments are performed on one NVIDIA Tesla V100 GPU with 16GB memory. For the training process of our DRNets, we select a learning rate in $\{0.0001, 0.0005, 0.001\}$ with Adam optimizer (Kingma & Ba, 2014), for all the experiments.

For baseline models, we followed their original configurations and further fine-tuned their hyper-parameters to saturate their performance on our tasks.

#### A.4.1 MULTI-MNIST-SUDOKU

For Multi-MNIST-Sudoku, we compared DRNets with CapsuleNet (Sabour et al., 2017) and ResNet (He et al., 2016). Because Sabour et al. (2017) did not provide a source code for CapsuleNet, we adopted the implementation of Laodar (2017), with minor modifications. For ResNet, we adopted a 18-layer ResNet architecture (Khanrc, 2017) to saturate its performance.

In Multi-MNIST-Sudoku, a data point corresponds to a $32 \times 32$ image of overlapping digits. For the optimization mode of DRNets, we generated $160,000$ input data points that all come from the *test set* of MNIST (LeCun et al., 1998) and every 16 data points form a 4-by-4 overlapping Sudokus. Thus, these $160,000$ data points form $10,000$ Sudokus. These $10,000$ Sudokus are used as the test set and shared across DRNets and baselines. For the generalization mode of DRNets, we split the training set of MNIST into three parts: $160,000$ data points for DRNets learning, $25,000$ *original MNIST images* for training conditional GAN and another $160,000$ data points for validation. Note that these three datasets are disjoint. Baselines share the same training set as the generalization mode of DRNets. By using the constraint-aware SGD, DRNet batches every 16 data points together, which forms an overlapping Sudoku as well as a maximal connected component in the *constraint graph*, to enforce the All-Different constraints among the cells of each Sudoku.

DRNet for Multi-MNIST-Sudoku: the encoder is made of two ResNet-18 models adapted from the PyTorch source code. The output layer for the first network has 8 dimensions, which models the two distributions $P_i$ and $Q_i$ for the two overlapping digits. Another network outputs eight 100-dimensional (800 dimensions in total) latent encoding $z_{i,j}$ to encode the shape of the possible eight digits conditioned on the input mixture, and is used by the generative decoder to generate the reconstructed digits. We use a conditional GAN (Mirza & Osindero, 2014) as our generative decoder, which is pre-trained using the digits in the *partial training set* (see the paragraph above) of MNIST. Note that this is the only supervision we have in this task, which is even weaker than the general concept of the *weakly-supervised setting* (Zhang et al., 2017). We adopted the implementation of Linder-Noren (2019) for our conditional GAN. On the other hand, the 10,000 overlapping Sudokus in the test set were all generated using the digits in the *test set* of MNIST, which had never been

| Method | Accuracy (%) | | Time |
|---|---|---|---|
| | Digit | Sudoku | |
| DRNets (Optimization w/ Restart) | **100.0** | **100.0** | 50min |
| DRNets (Optimization) | **99.9** | **98.6** | 28min |
| DRNets (Optimization w/o Reasoning) | 88.8 | 15.0 | 110min |
| DRNets (Generalization) | **98.0** | **75.7** | 13min+4hrs |
| CapsuleNet | 97.9 | 50.9 | 1min+30min |
| CapsuleNet + local search | 97.9 | 57.8 | 3hrs+30mins |
| ResNet-18 | 97.7 | 68.5 | 3min+10hrs |
| ResNet-18 + local search | 97.7 | 88.3 | 3hrs+10hrs |

Figure 9: Accuracy comparison. We show "test time + training time" for supervised baselines and the generalization mode of DRNet, and "solving time" for the optimization mode of DRNets.

seen, even by the conditional GAN. Moreover, we overlap the images of two digits pixel-wisely, maximizing the whiteness of the two images. For robustness concern, we used $L1$ loss as the reconstruction loss between the reconstructed mixture and the original input. For the initial weights, we set 0.01 for the cardinality constraints, 1.0 for the All-Different constraints, and 0.001 for the $L1$ loss. Finally, we trained DRNets for 100 epochs with a batch size of 100, and it took 50 minutes to finish the optimization and achieve the reported performance for the 10,000 overlapping Sudokus.

For the generalization mode of DRNets, we first "train" DRNets on the training set and validate its generalization performance on the validation set to apply the early stop mechanism. Finally, we start from the "trained" DRNets and further optimize it for 25 steps on the test set to achieve the reported performance. Note that, to generalize well on the test set, we "trained" DRNets for a longer time than the optimization mode. Essentially, the procedure of the generalization mode of DRNets is similar to standard supervised learning process except that we do not need labels to supervise DRNets. In contrast, DRNets are really "self-supervised" (Jing & Tian, 2019) by the Sudoku rules and the self-reconstruction, instead of the standard supervision by labeled data. Note that, during the test, instead of predicting the overlapping digits directly as other networks, we further optimize DRNets on the test set for 25 epochs to achieve a better result.

### A.4.2 CRYSTAL-STRUCTURE-PHASE-MAPPING

We illustrate DRNets for crystal structure phase mapping for two chemical systems: (1) a ternary **Al-Li-Fe** oxide system (Le Bras et al., 2014), which is theoretically-based, synthetically generated, with ground-truth solutions, and (2) a ternary **Bi-Cu-V** oxide system, which is a more challenging real system obtained from chemical experiments and is more noisy and uncertain. For each system, the input data points are mixtures of XRDs, associated with a composition graph identifying elemental compositions and the *constraint graph* of data points. Specifically, each XRD data point is associated with a 3-dimensional composition vector, which is the proportion of the three different metal elements at that data point (e.g., [80% of Al, 5% of Fe, 15% of Li]) and could help identify possible phases. Then, we can locate each data point into a triangular system. Note that, since the vector is a probability distribution, there are only 2 degrees of freedom and we can plot it in a 2-D triangle (See Fig.11). After locating each data point into the 2-D triangle as vertices, we did a Delaunay triangulation over those points to build edges among vertices. Therefore, we can use Breadth-First Search on this graph to sample paths in the composition graph and infer thermodynamic rules accordingly.

The XRD pattern of each data point is a $D$-dimensional vector representing the intensity of the mixture of XRDs at different diffraction angles (referred as $Q$ values). For Al-Li-Fe oxide system, we have 231 data points (mixtures of XRDs) in the composition graph, 159 stick patterns for the possible phases, and each data point has 650 different Q values $Q_i \in [15°, 80°]$ and the corresponding intensities $I_i \in [0, 1]$. For Bi-Cu-V oxide system, we have 353 data points in the composition graph, 100 stick patterns for the possible phases, and each data point has 4096 different Q values $Q_i \in [5°, 45°]$ and the corresponding intensities $I_i \in [0, 1]$. To better utilize the memory, we down-sampled the raw data of Bi-Cu-V oxide system to 512 different Q values. Note that, though we have hundreds of possible

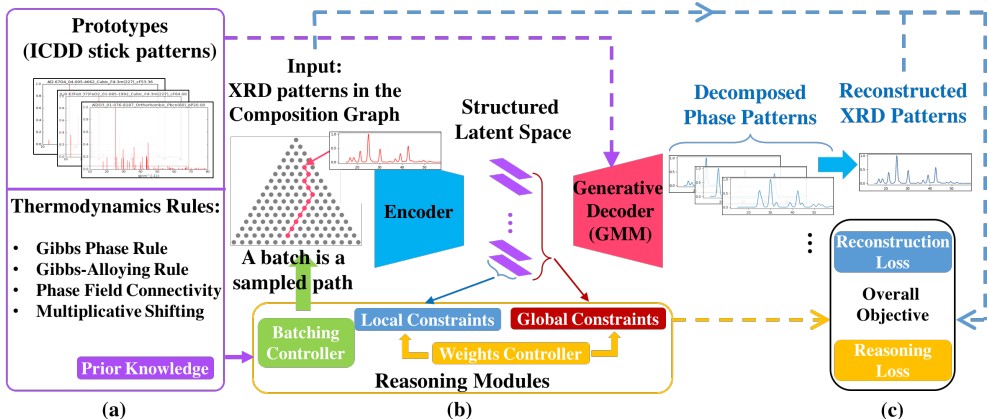

Figure 10: Deep reasoning networks (DRNets) for crystal-structure-phase-mapping. **(a)** Prior knowledge includes the ICDD stick patterns of possible pure phases, which are used to build the GMM generative module in the decoder, and the thermodynamic rules that help DRNets reason about the mixture of XRD patterns. **(b)** reasoning modules batch data points involved in a connected component of the *constraint graph* (a path in the composition graph) together, enforce that the structure of the latent space satisfies prior knowledge, and dynamically adjust the weights of constraints based on their satisfiability. **(c)** The overall objective combines responses from the generative decoder and the reasoning modules.

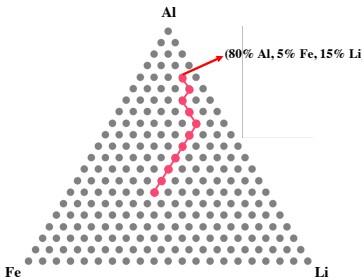

Figure 11: The composition graph of the Al-Fe-Li oxide system. The red path is a sampled path in the composition graph.

pure phases for each system, only a few phases would appear. For example, in Al-Fe-Li oxide system, only 6 of them appear and there are 15 different mixtures of those 6 pure phases exist in this system. For the Bi-Cu-V-O system, there are 13 pure phases and 19 different mixtures. Note that, each XRD data point is like a cell in the Multi-MNIST-Sudoku (with mixed pure phases) and each pure phase is like a digit. For Multi-MNIST-Sudoku, we know a priori that there are exact 2 digits in each cell but the number of mixed pure phases in each XRD is undetermined (1 to 3). Moreover, the number of possible candidate phases is way more than possible digits (e.g., 159 vs 8), which is the reason why this task is so challenging.

We also collected a library of possible crystal structures from the International Centre for Diffraction Data (ICDD) database. Each crystal structure (also named *phase*) is given as a list of diffraction peak location-amplitude pairs, (referred to as *stick pattern*), representing the ideal phase patterns measured in a perfect condition (see Fig.12). To model more realistic conditions, DRNets simulate the real phase patterns from *stick patterns* using Gaussian mixture models, where the relative peak locations and mixture coefficients are given by the stick locations and amplitudes. Moreover, the peak width, multiplicative location shift, and possible amplitude variance are parameterized by the latent encoding $z_{i,j}$ and used by the generative decoder to generate the corresponding possible phase patterns in the reconstructed XRD measurement.

Imposing thermodynamic rules is challenging, especially when constraints, such as *phase field connectivity* and *Gibbs-alloying rule*, potentially concern all data points in the composition graph.

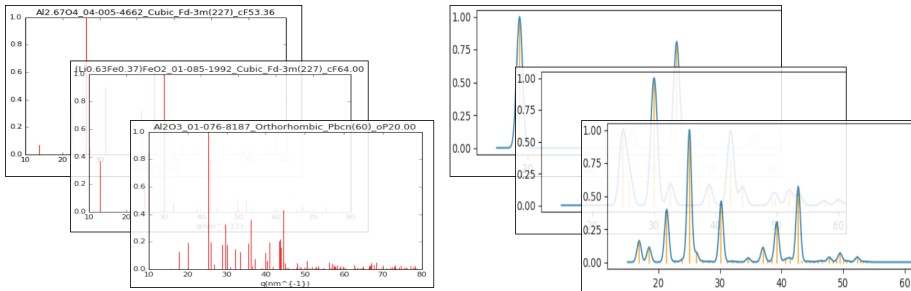

Figure 12: Some examples of stick patterns and their corresponding Gaussian Mixture Models. The horizontal axis denotes the Q values, and the vertical axis denotes the diffraction intensity.

In Multi-MNIST-Sudoku, where each overlapping Sudoku naturally forms the maximal connected components in the *constraint graph*, we can easily batch every 16 data points together to reason about the All-Different constraints among them. However, in Crystal-Structure-Phase-Mapping, since the maximal connected component involves all data points in the composition graph, neither batching all data points into the memory nor reasoning about the whole graph is tractable. Therefore, we devised a strategy of sampling the large connected component through many local structures (still connected components) and iteratively solve each of them. Specifically, for each oxide system, we sampled 100,000 paths in the composition graph via Breadth First Search to construct a path pool. Then, for every iteration, DRNets randomly sample a path from the pool and batches the data points along that path (see 10). Finally, we only reason about the thermodynamic rules along the path. By iteratively solving sampled local structures (paths) of the "large" maximal component, we can cost-efficiently approximate all global constraints.

We summarize the thermodynamic rules we imposed in DRNets:

**Gibbs Phase Rule:** This rule states the maximum number of co-existing phases, which is imposed via our relaxation of the k-sparsity constraints.

**Gibbs-Alloying Rule:** This rule states that if "alloying" happens, then the maximum number of possible co-existing phases should decrease by one. "Alloying" is a phenomenon that the stick locations of a phase (crystal structure) shift (change) along with adjacent data points. DRNet explicitly models the shifting ratio in the generative decoder and penalize the difference between adjacent data points along our sampled path. The reasoning module keeps track of the difference of shifting ratio between adjacent data points, and when it is larger than a threshold (0.001), we confirm the existence of "alloying" and reduce the maximum number of possible co-existing phases by one via adjusting the threshold $c$ in the k-Sparsity Constraints.

**Phase Field Connectivity:** This states that the distribution (also referred as activation) of a phase field should form a connected component in the composition graph, and the variation of the activation of each phase should also be smooth (see Fig.13). (Herein, the phase field refers to the co-existence of a combination of phases, including the existence of a pure phase.) We impose this rule by penalizing the difference of the phase distribution $P_i$ between adjacent data points along our sampled path.

**Multiplicative Shifting:** This states how a cubic crystal structure shifts when "alloying" happens, and this can also be used to approximate the shifting of other crystal structures. We explicitly modeled the multiplicative shifting in our generative decoder.

**Noise Threshold:** To remove negligible activations that are mainly caused by noise we applied simple post-processing that cuts-off all the activations that are lower than 1.0%.

Here, we visualized the DRNets' solution of Bi-Cu-V oxide system (see Fig.13 and the comparison among different methods Fig.14).

In our comparison, we evaluated the percentage of data points or phase field that satisfy each thermodynamic rule. Though IAFD enforced the thermodynamic rule using an external mixed-integer programming module, it may compromise some rules to achieve a better reconstruction error, which explains IAFD's result for Bi-Cu-V oxide system. The phase fidelity loss we mentioned in our comparison is the JS distance between the de-mixed pure phase and the closest ideal phase generated

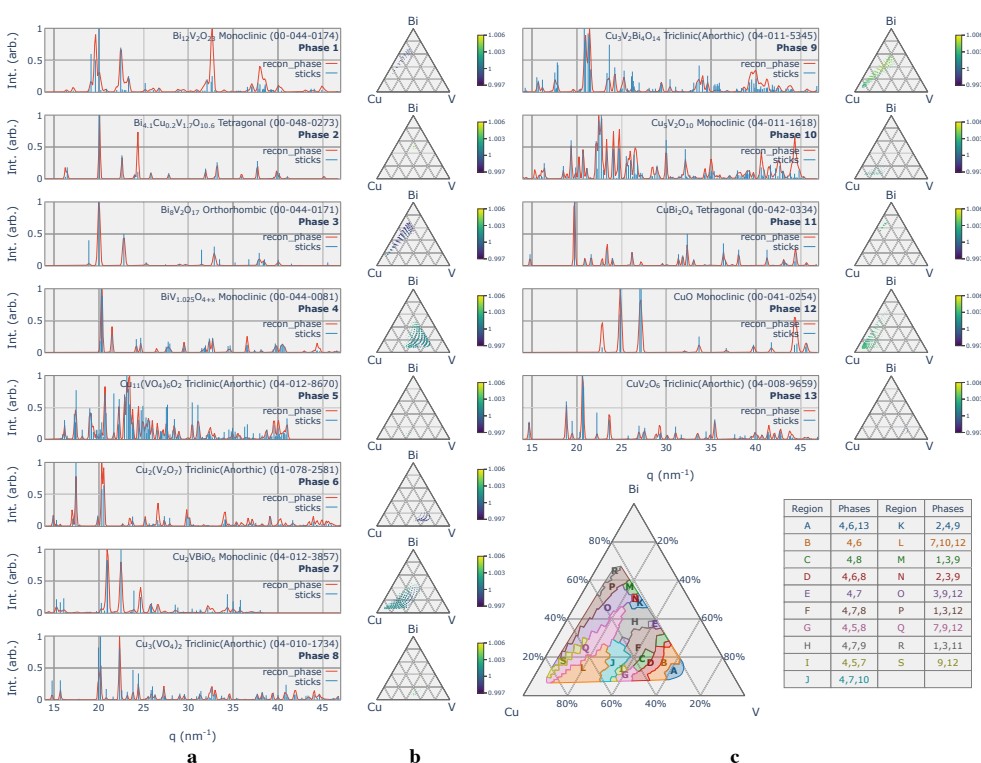

Figure 13: **DRNets' solution for the Bi-Cu-V oxide system.** **a.** The de-mixed crystal phases for the 353 XRD measurements of the Bi-Cu-V oxide system (each plot includes the signal for the recognized phase and the corresponding ICDD stick pattern). **b.** DRNets' phase concentration maps for the corresponding phases on the left of the map. Dot sizes are proportional to their estimated phase concentrations and heatmap denotes estimated shifting (alloying). **c.** DRNets' crystal phase map for the Bi-Cu-V-O system in the composition graph; the phase fields are labeled with corresponding crystal phases.

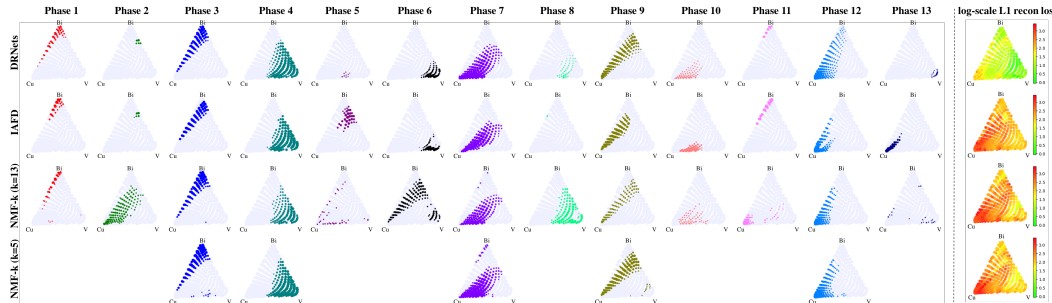

Figure 14: **Comparison of the activation map and the heatmap of L1 reconstruction loss for different methods for the Bi-Cu-V oxide system:** Each row denotes the activation of the different phases for the the different methods; Though we do not have ground truth for the Bi-Cu-V oxide system, the solution generated by DRNets satisfies all thermodynamic rules with excellent reconstruction performance; The heatmap on the right shows that DRNets reconstruct the XRD measurements much better than other methods with respect to the (log scale) L1 reconstruction under physical constraints of decomposed phases; In addition, materials science experts thoroughly checked DRNets' solution of Bi-Cu-V oxide system, approved it, and subsequently discovered a new material that is important for solar fuels technology.

using the ICDD stick patterns and the physical model proposed in Le Bras et al. (2014). The reason of using JS distance to measure the fidelity is that the location of peaks are the most important characteristics of a phase pattern. Therefore, by normalizing the XRD patterns of pure phases into probability distributions, we can use the JS distance to measure the mismatch of "peaks" between them.

In terms of the optimization process, DRNets took about 30 minutes to achieve the reported performance for both systems. IAFD and NMF-k have a similar time performance but a much worse performance w.r.t. the solution quality. In fact, for the Bi-Cu-V oxide system, both NMF-k's solution and IAFD's solution are not physically meaningful.

In summary, by combining reasoning and deep learning, DRNets significantly outperformed the state of the art and human experts on the crystal-Structure-Phase-Mapping instances, recovering more precise, interpretable, and physically meaningful crystal structure pattern decompositions, and even ***solving phase diagrams of chemical systems that had not been solved before***, such as the **Bi-Cu-V-O** system, but also other systems not reported here.

### A.4.3 OTHER EXPERIMENTS FOR COMBINATORIAL PROBLEMS

As a proof of concept of how DRNets can encode general combinatorial constraints using our entropy-based continuous relaxation, we solved 9-by-9 Sudoku puzzles and Boolean satisfiability problems (SAT) using DRNets. For those two tasks, we use a 3-layer-fully-connected network as our encoder and the reasoning modules.

| 5 | 3 |   |   | 7 |   |   |   |   |
|---|---|---|---|---|---|---|---|---|
| 6 |   |   | 1 | 9 | 5 |   |   |   |
|   | 9 | 8 |   |   |   |   | 6 |   |
| 8 |   |   |   | 6 |   |   |   | 3 |
| 4 |   |   | 8 |   | 3 |   |   | 1 |
| 7 |   |   |   | 2 |   |   |   | 6 |
|   | 6 |   |   |   |   | 2 | 8 |   |
|   |   |   | 4 | 1 | 9 |   |   | 5 |
|   |   |   |   | 8 |   |   | 7 | 9 |

| 5 | 3 | 4 | 6 | 7 | 8 | 9 | 1 | 2 |
|---|---|---|---|---|---|---|---|---|
| 6 | 7 | 2 | 1 | 9 | 5 | 3 | 4 | 8 |
| 1 | 9 | 8 | 3 | 4 | 2 | 5 | 6 | 7 |
| 8 | 5 | 9 | 7 | 6 | 1 | 4 | 2 | 3 |
| 4 | 2 | 6 | 8 | 5 | 3 | 7 | 9 | 1 |
| 7 | 1 | 3 | 9 | 2 | 4 | 8 | 5 | 6 |
| 9 | 6 | 1 | 5 | 3 | 7 | 2 | 8 | 4 |
| 2 | 8 | 7 | 4 | 1 | 9 | 6 | 3 | 5 |
| 3 | 4 | 5 | 2 | 8 | 6 | 1 | 7 | 9 |

Figure 15: A standard **9-by-9 Sudoku puzzle:** a partially filled Soduku has to be completed as a valid Sudoku.

For 9-by-9 Sudoku puzzles, we generated 10,000 instances using the dataset gathered by Gordon Royle (2014), where each Sudoku instance has 24 to 32 (uniformly distributed) known cells and is guaranteed to have one unique solution (e.g., see Fig.15). Because a standard 9x9 Sudoku puzzle requires reasoning about the unknown structure based on given clues, we need to treat each entire Sudoku as a single input data point. Therefore, in this task, even the All-Different constraints are conceptually the local constraints since each of them only concerns a single data point. We used a one-hot encoding for digits 1 to 9 and the empty cell (denoted as 0), and the entire Sudoku is an 810-dimensional input data. We used a 3-layer-fully-connected network with batch normalization (Ioffe & Szegedy, 2015) as the encoder, where every hidden layer has 2048 units and the output is an 81-by-9 matrix, which represents the digit distributions (1 to 9) for 81 cells. Moreover, we enforced the distribution of every known cell to collapse to the digit in that cell. For the initial weights, we set 0.0001 for the cardinality constraints and 1.0 for the All-Different constraints. Finally, we trained DRNets for 800,000 iterations with a batch size of 500, and it took 1 hour to solve the 10,000 9x9 Sudokus with the accuracy reported in this paper.

In our experiments, DRNets achieved the same level of performance as the Recurrent Relational Networks (RRNets) (Palm et al., 2017), which is the state-of-the-art supervised deep learning 9x9 Sudoku solver (see Table 1).

| Instances (10,000) | DRNets | DRNets + Restart | NeuralSAT | PDP | RRNets |
|---|---|---|---|---|---|
| 3-SAT n=30 m=129 | 81.0% (4min) | **99.0%** (33min) | 45.5% (2min+1hr) | 78.9% (5min+2hr) | NA |
| 3-SAT n=50 m=215 | 63.3% (7min) | **94.0%** (47min) | 26.1% (3min+1hr) | 62.2% (8min+2hr) | NA |
| 3-SAT n=100 m=430 | 34.7% (17min) | **77.9%** (2hr) | 4.7% (5min+1hr) | 31.4% (2hr+2hr) | NA |
| 3-SAT n=30, m=90 | 97.9% (5min) | **99.9%** (6min) | 78.5% (2min + 1hr) | 99.1% (4min + 2hr) | NA |
| 3-SAT n=50, m=150 | 98.2% (7min) | **99.4%** (8min) | 70.1% (3min + 1hr) | 99.2% (7min + 2hr) | NA |
| 3-SAT n=100, m=300 | 98.1% (20min) | **99.7%** (22min) | 52.9% (5min + 1hr) | 99.1% (2hr + 2hr) | NA |
| 9x9 Sudoku | 99.5% (1hr) | **99.8%** (1hr) | NA | NA | 99.6% (1min+1day) |

Table 1: Percentage of instances solved for 3-SAT ($m/n = 4.3$ and $m/n = 3.0$) and standard 9x9 Sudoku (24 to 32 known cells). We show the "test time + training time" for supervised baselines and the "solving time" for our unsupervised DRNets. The units min, hr, day denote minute(s), hour(s) and day(s). $m, n$ denote the number of literals and clauses, respectively. NA, not applicable. DRNets, without supervision, outperform the supervised state of the art.

For SAT problems, we generated 10,000 satisfiable random 3-SAT instances of different difficulties based on the number of literals $n$ and the number of clauses $m$, and our goal is to find a valid assignment for each literal. We challenged our DRNet with the hardest random 3-SAT instances, where #clauses/#literals=4.3 (Mitchell et al., 1992), i.e., $n = 30, m = 129$, $n = 50, m = 215$ and $n = 100, m = 430$. For easier instances (e.g. #clauses/#literals = 3.0), DRNets can almost solve all instances (see Table 1).

We use a 3-layer-fully-connected network as the encoder, where the number of hidden units in the network is 2048, 2048, 2048. We used the standard CNF representation of 3-SAT as the input data, so that each data point is an $m$-by-3 matrix and the three values in the $j$-th row represent the three literals in the $j$-th clauses. For the initial weights, we select a value from {0.05, 0.03, 0.025, 0.02, 0.01} to be the weight of the entropy loss as we described in the Fig.4 of the main paper. For the three settings of different difficulty, we consistently trained DRNets with a batch size of 100 and the running time for solving 10,000 instances varies from several minutes to a couple of hours.

We compared DRNets with NeuroSAT Selsam et al. (2018) and PDP (Amizadeh et al., 2019). Both NeuroSAT and PDP are the state-of-the-art deep learning SAT solvers with one-bit supervision. In addition, PDP needs extra optimizing process to solve SAT instances during the test phase, where it also applied the *restart* mechanism in their framework. For fair comparison, we saturated the performance of all our baseline models. For all instances, DRNets took less than 2 hours to achieve the reported performance with the *restart* mechanism. Without supervision, DRNets outperformed both supervised baseline models.

Interestingly, though DRNets are best suited for problems that combine deep learning and reasoning, such as de-mixing Multi-MNIST-Sudokus or crystal structure phase mapping, it still achieved such a promising result in pure combinatorial problems. These results further demonstrate that DRNets can encode a broad range of combinatorial constraints and prior knowledge and effectively combine deep learning with reasoning.

