# OpenReview forum: "Deep Reasoning Networks:  Thinking Fast and Slow, for Pattern De-mixing"
_ICLR.cc/2020/Conference — Reject_

### Official Review · AnonReviewer3 · 2019-10-23
**Official Blind Review #3**

**Rating:** 6

**Review:**

This paper proposes a new encoder-decoder framework that combines prior knowledge-based regularization and constrained reconstruction for unsupervised and weakly-supervised classification in structure rich scenarios. This framework injects prior knowledge in the form of relaxed constraints that act as regularization during the training of the encoder network. Some of the constraints concern sets of training examples. In this case, the paper proposes corresponding sampling schemes. Three experiments demonstrate the efficacy of the model. The first is a synthetically created 4x4 Sudoku made of overlaid MNIST digits. The other two are based on predicting crystal structures from x-ray diffraction measurements. Here, the first experiment is on simulated data for the Al-Li-Fe oxide system, while the other is performed on real measurements for the Bi-Cu-V oxide system.

Overall, I believe that the proposed framework could be a significant contribution to the fields of representation learning and constrained optimization. However, the paper exhibits serious shortcomings, which require revision.

First, the positive aspects of the paper:
•	The framework is simple yet ingenious. It makes intelligent use of constraints in the form of regularization to guide the training of the encoder. Furthermore, it enables the direct design of the latent representation through the use of (pre-trained) generative models for constrained reconstruction of data points.
•	The proposed entropy-based method for relaxation of discrete constraints is intuitive and potentially adaptable for further constraints.
•	The experiments presented in this paper are well chosen. They demonstrate the contribution of the model to both general CV data as well as a specialized domain, where it can solve both simulated and real scenarios.
•	The paper provides an extensive literature survey, which makes it easy to embed the presented work in the proper context. However, I propose to remove the paragraph titled “Other less closely related work” as the connection to the current work is not clear, and the space could be used more effectively (see below).

Unfortunately, this paper has a couple of major flaws:
•	The results for DRNets (Generalization) on the MNIST Sudoku are compromised because the model trained on the test set for 25 epochs after being trained on the training set. Honestly, I was baffled to read the following sentence in the appendix: “Note that, during the test, instead of predicting the overlapping digits directly as other networks, we further optimize DRNets on the test set for 25 epochs to achieve a better result.” What is more, the main paper does not even mention this fact!
•	Although this paper relies on empirical verification of its proposition, the experimental results are almost impossible to interpret with just the information provided in the paper. Both experiments are poorly described, and even after reading the appendix several times, some serious detective work was necessary to piece together what happened in the experiments. The XRD experiments are especially hard to decipher, even with a physics background. Many vital components remain shrouded in mystery: What is a composition graph, and how are the paths sampled from it? How does the restart method, which is part of the results, work? Why are only six phases shown in the phase concentration visualizations if there were 159 possible phases? Are these the first six, a random subset, or were the other phases not realized?
•	The paper introduces the constraint-aware SGD algorithm to incorporate batching rules into the training of the encoder. On pages 2 and 6 and in Algorithm 1, I found the statement that the weights for each constrained are updated dynamically. However, that is where the information on the dynamic update method ends. Nowhere in the paper or the appendix did I find an explanation of how this is done. As this mechanism is a critical component of the proposed framework, the absence of an explanation is a significant oversight.
Other remarks:
•	In the context of global constraints, the paper talks about a constraint graph. If I understand the creation of this graph correctly, this graph has several connected components in which every element connects to every other element. As such, this seems to be a collection of sets rather than a real graph. This is especially confusing in the case of XRD, where all data points are in the same global constraint, leading to a fully connected “graph”.
•	Although I appreciate the reference to Kahneman’s model of the mind, I suggest to remove the first two paragraphs from the introduction and use the space to motivate the de-mixing problem instead. While it is a compelling (but not novel) observation, the analogy to system 1 and system 2 does not benefit the proposed work in the slightest.
•	In general, I fail to see the connection between reasoning and the proposed work. The model itself is an encoder-decoder network that cannot reason. It does not discover any new rules during training. All the reasoning has to be done manually beforehand to be then incorporated in the form of constraints. To clarify, I do believe that there is value in the presented work, but not necessarily in the way, it is advertised.


**Experience Assessment:**

I have published one or two papers in this area.

**Review Assessment: Checking Correctness Of Derivations And Theory:**

I assessed the sensibility of the derivations and theory.

**Review Assessment: Checking Correctness Of Experiments:**

I carefully checked the experiments.

**Review Assessment: Thoroughness In Paper Reading:**

I read the paper thoroughly.

---

> ### Author Response · Authors · 2019-11-07
> **Details of experiments and clarification of the "generalization mode"**
>
> Thank you so much for appreciating our work!
>
> Indeed, we also included the performance of DRNets on some pure NP-C problems including original Sudoku problems and 3-SAT problems. You can check them in the last two pages of the appendix.
>
> Thank you for pointing out the "redundant" part of the related work. We will consider erasing that paragraph to save space for other content.
>
> 1. Thank you for pointing out the confusion of the "generalization mode" of DRNets. In fact, DRNets mainly target on "solving" unsupervised pattern de-mixing tasks, instead of generalization. Because in real tasks, such as crystal-structure phase mapping, we only have hundreds of data points, which is not enough to do any generalization.
> However, in Multi-MNIST-Sudoku, we found that by solving enough instances together, the network naturally could generalize to unseen instances. This phenomenon resembles a "self-learning" process, where we can actually learn a model without labels if we have enough unlabeled data points. Therefore, we think it is interesting to show this phenomenon in our paper. However, limited by space, we are sorry that we didn't address this properly. We do extra 25-optimization steps for improving the generalization performance on unseen instances, which increases the one-shot Sudoku accuracy from 50.5% to 75.7%. We use the validation set for determining the best # of extra optimizations. We will clarify this in the paper.
>
> 2. We do appreciate that you read our paper thoroughly. To be honest, the crystal-structure phase mapping is a real-world task, which includes a lot of domain knowledge and details. Therefore, we proposed this Multi-MNIST-Sudoku task as a glimpse of the crystal-structure phase mapping.  We are sorry for the confusing terminology in our description and we will include more details of this experiment to make it reproducible for other readers. In fact, we have organized our code as well as the dataset into a runnable package, and we will release it with a document for people to fully understand our code.
>
> For your specific questions:
> (1) composition graph: Each XRD data point is associated with a 3-dimensional vector, denoting the percentage of 3 elements at that point (e.g., [a% of Li, b% of Fe, c% of Al]). Then you can locate each data point into the triangular system (note that, the vector is a probability distribution so that there are only 2 degrees of freedom and it can be plotted in a 2-D triangle.) Visually, we will have a composition graph like the triangle plot in Fig.10 (appendix). After locating each data point into the 2-D triangle as vertices, we did a Delaunay triangulation over those points to build edges among vertices. Finally, we did a Breadth-First Search on this graph to sample paths.
>
> (2) restart mechanism for Multi-MNIST: Since DRNets directly incorporate logical constraints, we can check whether those constraints are satisfied at the end of a run. If not, for instances with violated constraints, we re-run the algorithm again on them. We didn’t restart for crystal-structure phase mapping.
>
> (3) 6 phases out of 159 possible phases: Though there are 159 possible pure phases for the Al-Fe-Li-O system, only 6 of them appear and there are 15 mixtures of those 6 pure phases exist in this system. For the Bi-Cu-V-O system, there are 13 pure phases and 19 different mixtures. Note that, each XRD data point is like a cell in the Multi-MNIST-Sudoku (with mixed pure phases) and each pure phase is like a digit. For Multi-MNIST-Sudoku, we know a priori that there are exact 2 digits in each cell but the number of mixed pure phases in each XRD is from 1 to 3. Moreover, the number of possible candidate phases is way more than possible digits (e.g., 159 vs 10), which is the reason why this task is so challenging. We will make this point more clear in our final version.
>
> 3. Dynamic update of penalty weights: please check similar response to reviewer #1(Q1)
>
> 4. The constraint graph shows how data points are linked through different constraints. Yes, you can think of it as a collection of sets when we can batch each maximal connected component together. When it comes to the case of XRD, the graph is fully-connected due to global constraints. Therefore, we no longer batch the maximal connected component together but sample a path in this graph to reason about a local structure of those global constraints. Though the constraint graph is a fully-connected graph, we prefer to sample paths in the composition graph, given it is easier to reason about those thermodynamic rules on a path of composition graph.
>
> 5. The reasoning does happen in DRNets: See comments to reviewer #2
>
> Thanks again! We believe this framework is indeed quite “ingenious :-)!” expandable for other constraints and domains, very relevant for scientific unsupervised tasks with rich prior-knowledge.  We will improve the description of our problem domains and we hope you champion this paper!

---

### Official Review · AnonReviewer1 · 2019-10-25
**Official Blind Review #1**

**Rating:** 3

**Review:**

This work proposes a framework for solving de-mixing problems. The hard constraints from human inputs about a specific problem are relaxed into continuous constraints (the "slow" reasoning part), and a reconstruction loss measures the fitness of the inferred labels with the observations (the "fast" pattern recognition part). Due to the relaxation inference becomes an optimization problem, and on a Sudoku task and a crystal-structure-phase-mapping recovery task (both de-mixing tasks), the proposed method gets very good performance (100% for all Sudoku tasks including one in the appendix).

Pros:
1. The method works well for the two demixing tasks.
2. It "led to the discovery of a new material that is important for solar fuels technology"

Cons:
1. The generative decoder seems to be pretrained on both tasks instead of learned (correct me if I misunderstood), and I'm not sure if this approach can work in cases where we don't have access to such a generative decoder, so branding the approach "deep reasoning network" might be an overclaim.
2. No reasonable baselines are used: The supervised baseline in Sudoku does not use those handcrafted constraints at all. Given pretrained decoders, a reasonable baseline would be randomized optimization methods such as simulated annealing, which might also solve the two tasks listed here.
3. This paper proposes a deep reasoning framework with relaxation and continuous optimization, but it is unclear whether this can solve general reasoning problems such as multi-hop QA or some NP-hard integer programming problems.

Questions:
1. In algorithm 1, how are the penalty weights and thresholds adjusted?
2. How to determine whether a run needs to restart?

Overall this work points an interesting direction of combining reasoning and pattern recognition in the same network and the proposal works well on two de-mixing problems. However, I am not convinced that the proposed solution can generalize to tasks other than the tasks proposed here, and the usage of pretrained generative decoders undermines the significance of this work. Therefore, I am inclined to reject this paper.


---updates after reading authors' rebuttal----
Thanks for revising the paper and addressing my concerns! However, my concern Con #2 has not been fully addressed. I think a reasonable baseline (at least for Sudoku) is simulated annealing, such as in https://www.researchgate.net/publication/220704743_Sudoku_Using_Parallel_Simulated_Annealing. I believe that with restarts those baselines would also solve the Sudoku problem.

Another concern I still have is the claim of "reasoning", and I'd suggest to narrow down the claim to be only on pattern de-mixing, since the reasoning part seems to be writing down continuous constraints from the discrete constraints (same as the concern in review #3). Although the proposed approach can solve some NP-C integer programming problems, it is unclear based on the experiments here whether it can work for general reasoning tasks (e.g., DROP https://allennlp.org/drop or listops https://arxiv.org/pdf/1804.06028.pdf) without writing new rules manually.

Besides, after reading Reviewer 3's comments, I also feel it unsuitable to train DRNet (generalization) on test set for 25 epochs even though you made it explicit in the revised paper. I'd recommend removing that experiment since it doesn't change this work that much.


**Experience Assessment:**

I do not know much about this area.

**Review Assessment: Checking Correctness Of Derivations And Theory:**

I carefully checked the derivations and theory.

**Review Assessment: Checking Correctness Of Experiments:**

I carefully checked the experiments.

**Review Assessment: Thoroughness In Paper Reading:**

I read the paper thoroughly.

---

> ### Author Response · Authors · 2019-11-07
> **DRNets are general and our baselines are reasonable**
>
> “Pros:
> 1. The method works well for the two de-mixing tasks.
> 2. It "led to the discovery of a new material that is important for solar fuels technology"
>
> These are not small achievements :-)!
>
> Con#1: The generative decoders are either a pre-trained or a parametric model (GMM), which is learned/obtained using prior knowledge. For example, we assume that we have prototypes of single hand-written digits in Multi-MNIST-Sudoku and we have the ICDD database to provide pure phase patterns. Note  DRNets target unsupervised/weakly-supervised pattern de-mixing tasks, where we don’t have ground-truth labels for the de-mixed patterns. Therefore, without any prior knowledge of what single patterns may look like, the de-mixing tasks are ill-posed and intractable. Indeed, such prior knowledge is necessary even for human experts to solve those tasks. Thus, having access to the prior knowledge of possible single patterns, which can be used to build the generative decoder, is a very practical and reasonable assumption for unsupervised/weakly-supervised de-mixing tasks.
>
> Please see the explanation to reviewer#2 about DRNets’ reasoning system.
>
> Con#2: The Multi-MNIST-Sudoku experiment is just a simplified example of the challenging task — crystal-structure phase mapping. Our comparison is to show how we can boost pure deep learning models by incorporating prior knowledge such as rules and constraints. We compared our unsupervised methods with state-of-the-art supervised approaches. Thus, the baseline models have label supervision while DRNets “reason” about rules. We thought that it was a fair comparison. Moreover,  we also tried your suggestion of using Sudoku rules for ResNet and CapsuleNet. Specifically, we did a local search for the top-2 most likely digits (one from 1-4 and another one from 5-8) for each Sudoku of the two overlapping Sudokus and try to satisfy Sudoku rules with minimal modification compared with the original prediction. The process took about 3 hours (we can't try top-3, it takes too long) for each model and increased their Sudoku accuracy from 68.5% -> 88.3% (ResNet) and 50.9% -> 57.8% (CapsuleNet). Though imposing Sudoku rules for these baselines could improve their performance, the barrier between the deep learning model and the post-process reasoning (local search) makes their performance far from DRNets' 100% accuracy. This further confirmed the advantage of combining deep learning and logical reasoning seamlessly and we are happy to include this in our paper. Thanks for the great suggestion.
>
> For crystal-structure phase mapping, we have compared our model with the state-of-the-art approaches in this area, i.e., NMF-k and IAFD, where IAFD has directly incorporated all constraints as we have and NMF-k incorporated those constraints indirectly. Therefore, these two models are the best baselines we can have so far.
>
>
> Q1: We initialize penalty weights and thresholds for penalty functions using hyper-parameters. During training, we check the satisfiability of constraints after several epochs and increase the penalty for violated constraints. This mechanism is mainly designed for Crystal-Structure Phase Mapping because DRNets can already achieve perfect performance for Multi-MNIST-Sudoku with fixed weights. For example, the threshold c of k-sparsity is initialized as logk, which is the entropy of the case that the probability mass is evenly distributed among k entities. Thus, it could be the case that there are more than k entities, but their probability mass is not evenly distributed. Hence, we check the satisfiability of k-sparsity constraint: if the entropy is already below the current threshold (logk) and there are still more than k entities with probability mass more than epsilon (0.01), we decrease the threshold c to keep enforcing the model to minimize the entropy to reach the k-sparsity.
>
> Q2:
> Since DRNets directly incorporate logical constraints, we can check whether those constraints are satisfied at the end of a run. For the instances violating constraints, we re-run the algorithm again.
>
> Con#3 and “I am not convinced that the proposed solution can generalize to tasks other than the tasks proposed here”:
> Indeed, the DRNet framework is general and we do have some results for pure NP-C problems such as original Sudoku (9x9)  and 3-SAT problems. Please check the appendix, where we show that DRNets outperform existing state-of-the-art deep learning methods for those pure NP-C problems. For example, DRNets solve 3-SAT, 100 literals and 430 clauses, which is above the capability of existing deep learning methods. Note, however,  DRNets are mainly meant for problems that require combining the pattern recognition and logical reasoning instead of pure reasoning problems.
>
> DRNets are relevant to other scientific tasks and even other domains. Also, as you recognize, our results are very good for these tasks, leading even to new discoveries. We hope you re-consider our score. Thanks!

---

### Official Review · AnonReviewer2 · 2019-10-30
**Official Blind Review #2**

**Rating:** 3

**Review:**

This paper introduces a deep reasoning networks for de-mixing overlapping patterns with some logic constraints. There are two applications considered in the paper: de-mixing overlapping hand-written digits and inferring crystal structures of materials from X-ray diffraction data. The experiments indicate the proposed method work pretty well on these tasks.

I like the general idea of this paper, since it has the flavor of combining deep learning with logic rules, although I feel weird to view the generative decoder as thinking fast and the reasoning modules as thinking slow. The notion of thinking fast and slow in the model does not well match the intuition given in the first paragraph of the introduction. The so-called reasoning module is essentially some contraints (i.e., regualrization losses) and a training data sampler. It is far away from the concept of (symbolic or logic) reasoning. There is not too much reasoning happening here. The way the paper relaxes the discrete logic constraints to continuous and differentiable objective that can be jointly optimized by SGD is interesting, which is similar to [Harnessing deep neural networks with logic rules, ACL 2016]. The carefully designed training data sampler that samples data according to a constraint graph also resembles GraphRNN, as the authors have mentioned in the paper. I feel the combination of these techniques is definitely interesting but also somehow incremental. I am not a big fan of some big claims in the paper. The reasoning modules are not what I expect.

For the experiments, I think the authors do a good job presenting these experimental details and evaluations. These experiments are interesting and also show some advantages of the propose method. However, some baselines are also doing pretty well, indicating that the task is not difficult in general.

**Experience Assessment:**

I do not know much about this area.

**Review Assessment: Checking Correctness Of Derivations And Theory:**

I assessed the sensibility of the derivations and theory.

**Review Assessment: Checking Correctness Of Experiments:**

I assessed the sensibility of the experiments.

**Review Assessment: Thoroughness In Paper Reading:**

I read the paper at least twice and used my best judgement in assessing the paper.

---

> ### Author Response · Authors · 2019-11-07
> **Contribution beyond incremental and reasoning**
>
> Thank you for your review. Yes, we "stand on the shoulders of giants," but what makes our framework novel (not incremental) is the unique combination of ideas to deal with challenging unsupervised or weakly-supervised pattern de-mixing problems.
>
> "It is far away from the concept of (symbolic or logic) reasoning.” We disagree with this comment (see e.g, https://en.wikipedia.org/wiki/Reasoning_system). Reasoning system: given a set of axioms and rules, an inference procedure computes what follows. For example, in standard Sudoku, the inference procedure finds the values for missing cells.  In logic, the axioms and rules can be written using propositional logic and the inference engine can be e.g., resolution or resolution and search, etc. Another example of a reasoning system is a constraint solver. In our case, we show how to encode the problem as a constraint optimization problem, encoding the Sudoku (and phase mapping ) rules using entropy-based functions and then use Lagrangian relaxation, and constraint-aware SGD to do the reasoning. In the appendix, we show that DRNets can solve standard Sudokus to further illustrate how DRNets can infer (“reason about”) the missing values. But DRNets also reason about Sudoku rules (and thermodynamic rules) to make sense of the noise input patterns. A typical logical reasoning system cannot reason about noisy data: in contrast, the strength of deep reasoning is to be able to make sense of noisy patterns. That is a key novelty of our approach – we combine pattern recognition using deep learning with explicit constraint reasoning about rules, encoded through entropy-based functions and Lagrangian relaxation plus constraint-aware SGD. This combination of so many ideas makes our framework very powerful.
>
> "Harnessing deep neural networks with logic rules, ACL 2016" also utilizes logic rules to enhance deep learning. Good point. However, their framework is totally different from ours.  We didn't compare with them because their framework is only applicable to **supervised** settings, where they have massive labeled data points as their main supervision, which diminishes the role of logic rules.  In their experiments, the logic rules only increase their performance by less than 0.5%. Again, note that in contrast we are mainly interested in
> unsupervised/weakly-supervised settings for which we do not have labels for the mixtures, but we have only prototypes or idealized versions of what the digits or (phases) look like - for example, for the phase mapping not only do we only have unlabeled mixture data we also
> only have a few data points (<500).  The MNIST is not a good representative of these challenges since we can generate a large number of Sudokus (10,000). Also, in the MNIST we deal with 10 digits while in the phase mapping we can have possibly hundreds of pure phases (e.g., 159 for Al-Fe-Li). Furthermore, in the MNIST setting, we assume that there are always two overlapping digits: in the phase mapping, we do not know a priori how many phases are overlapped, which increases the combinatorial complexity of the problem and makes it even more important that the system could reason about the underlying thermodynamic rules.
>
> Thanks for appreciating our experiments.
>
> “However, some baselines are also doing pretty well, indicating that the task is not difficult in general”: The baselines actually perform poorly in both tasks: In Multi-MNIST-Sudoku, the baselines could only recover less than 70% overlapping Sudokus even with the supervision of labeled data points (which DRNets don’t have). In contrast, DRNets (with the supervision of the rules) recover 100% Sudoku puzzles without any labeled data. The difference in crystal-structure phase mapping is even more significant. For the Al-Fe-Li system, the phase diagram (Fig.7) recovered by baselines are far from the ground-truth while DRNets perfectly recovered it. For the Bi-Cu-V system, none of the baselines could generate a meaningful solution (the phases discovered by both IAFD and NMF-k are far from real phases (huge phase-fidelity loss); the solutions from both IAFD and NMF-k break the thermodynamic rules) and as far as we know DRNet is the first model that can solve this system.
>
> "I am not a big fan of some big claims in the paper." We understand your comment. But at the same time, there are advantages of seeing the "big picture". In fact, thinking of this framework as a general reasoning framework expanded our horizons and was very helpful for the students and senior researchers to have a broader perspective of its possibilities. For example, that led to the phase mapping application and we are now applying it to other problems in seemingly different domains such as species distributions for which there is prior knowledge about constraints on the species interactions and other settings.
>
> Thanks again, please let us know if you have additional questions.

---

### Author Response · Authors · 2019-11-07
**Revised version [minor revisions] and highlights of main contributions from reviewers.**

Dear Reviewers,

Thanks again for your time and comments.

We submitted a revised version addressing your comments and concerns.

Here is a list of the changes, per your suggestions:

(0) Changed the title to deep reasoning network for unsupervised pattern de-mixing

(1) Changed intro paragraph.

(2) Removed “less closely related work” paragraph.

(3) Added an explanation of what a reasoning system is in the first paragraph of section

     "Analogously, in a reasoning system,  an inference procedure derives what follows from an initial set of axioms and rules. For example, in a standard 9x9 Sudoku, an inference procedure identifies the missing cell values of the input Sudoku. A constraint solver is a particular type of reasoning system in which axioms and rules are expressed as constraints and the inference procedure is a search method.”


(4) Added the comparison between DRNets to supervised models + local search to enforce Sudoku Rules (both the description and the table in Fig. 5).

     "To saturate the performance of baseline models, we also applied a post-process local search for them to incorporate the Sudoku Rules. Specifically, we did a local search for the top-2 (top-3 would take too long to search) most likely choice of digits for each Sudoku of the two overlapping Sudokus and try to satisfy Sudoku rules with minimal modification compared with the original prediction."

(5) Added a reconstruction loss heatmap to the Fig.7 to emphasize the different performance between DRNets and other baselines. Note how much better (greener) DRNets reconstruct the input spectrograms.

In the appendix:

(6) Added detailed description of how to dynamically adjust the weights and the thresholds of penalty functions. [A.2, last paragraph of page 12.]

(7) Added detailed description of how to apply the restart mechanism. [A.3 page 13]

(8) Added detailed description and a figure to illustrate what a composition graph is. [A.4.2, first two paragraphs, page 14; Fig.11]

(9) Added plots of the Bi-Cu-V oxide system to help people better understand the XRD phase mapping task. [Fig.13; Fig.14]


Also, thank you for pointing out the many positive and exciting aspects of this paper!

Reviewer #3:  Your expertise: “I have published one or two papers…” You are the most knowledgeable of the group.  Your points are right on target concerning the potential and generality of DRNets! Please champion our paper!

Here are some highlights from Reviewer #3:

·      “The framework is simple yet ingenious.”
·      “It makes intelligent use of constraints in the form of regularization to guide the training of the encoder.”
·      “Furthermore, it enables the direct design of the latent representation through the use of (pre-trained) generative models for constrained reconstruction of data points.”
·      “The proposed entropy-based method for relaxation of discrete constraints is intuitive and potentially adaptable for further constraints. “
·      “The experiments presented in this paper are well chosen. They demonstrate the contribution of the model to both general CV data as well as a specialized domain, where it can solve both simulated and real scenarios. “
·      “The paper provides an extensive literature survey, which makes it easy to embed the presented work in the proper context.


Reviewer #1 and #2:  Your expertise: “I do not know much about this area.”  We appreciate the time you put into the review and careful reading of the paper. You raise some valid points and point out very positive points about our paper. For example, the challenge of problems concerning crystal phase mapping that was way beyond the capabilities of human experts is in itself a major achievement! In fact, that is how our collaborators were able to discover new material for solar fuel technology. Nevertheless, we do feel that you underestimated the contributions of our paper and you were very harsh with your score. Please re-consider it and discuss it with reviewer#3!

Highlights from Reviewer #1 and #2:
·      “The method works well for the two demixing tasks. “
·      “ It "led to the discovery of a new material that is important for solar fuels technology"
·      “I like the general idea of this paper, since it has the flavor of combining deep learning with logic rules”

To all: we are very proud of this work: It led to the discovery of new solar fuels and solved other unsolved crystal structure phase mapping problems. Phase mapping is a truly challenging problem. The framework is general and we believe it has a lot of potential for a variety of applications.  We are actually applying it to generate 3D reconstructions of 2D images (e.g., for medical applications, to reconstruct bones from medical images) using medical prior knowledge and reasoning about the shape of the bones and body anatomy.  We quickly edited the paper to address your concerns but of course, we are happy to further improve it based on your feedback. Thanks!

---

### Decision · Program_Chairs · 2019-12-19

**Decision:**

Reject

**Comment:**

The paper received mixed reviews of WR (R1), WR (R2) and WA (R3). AC has carefully read all the reviews/rebuttal/comments and examined the paper. AC agrees with R1 and R2's concerns, specifically around overclaiming around reasoning. Also AC was unnerved, as was R2 and R3, by the notion of continuing to train on the test set (and found the rebuttal unconvincing on this point). Overall, the AC feels this paper cannot be accepted. The authors should remove the unsupported/overly bold claims in their paper and incorporate the constructive suggestions from the reviewers in a revised version of the paper.